# No Significant Association between 25-OH Vitamin D Status and SARS-CoV-2 Antibody Response after COVID-19 Vaccination in Nursing Home Residents and Staff

**DOI:** 10.3390/vaccines11081343

**Published:** 2023-08-08

**Authors:** Eline Meyers, Evelien De Smet, Hanne Vercruysse, Steven Callens, Elizaveta Padalko, Stefan Heytens, Linos Vandekerckhove, Piet Cools, Wojciech Witkowski

**Affiliations:** 1Department of Diagnostic Sciences, Faculty of Medicine and Health Sciences, Ghent University, 9000 Ghent, Belgium; eline.meyers@ugent.be (E.M.);; 2HIV Cure Research Center, Department of Internal Medicine and Pediatrics, Ghent University Hospital, Ghent University, 9000 Ghent, Belgium; 3Research and Analytics, Liantis, 8000 Bruges, Belgium; 4Department of Internal Medicine and Pediatrics, Ghent University Hospital, Ghent University, 9000 Ghent, Belgium; 5Department of Medical Microbiology, Ghent University Hospital, 9000 Ghent, Belgium; 6Department of Public Health and Primary Care, Faculty of Medicine and Health Sciences, Ghent University, 9000 Ghent, Belgium

**Keywords:** vitamin D, 25-hydroxyvitamin D, older adults, nursing home residents, nursing homes, SARS-CoV-2, COVID-19 vaccination

## Abstract

Vitamin D is an essential nutrient for various physiological functions, including immunity. While it has been suggested that higher vitamin D levels/supplementation are associated with a better immune response to COVID-19 vaccination, conflicting data exist. Therefore, we aimed to investigate the association between vitamin D (25-hydroxyvitamin D) deficiency/supplementation, and SARS-CoV-2 antibody responses post-vaccination in nursing home residents (NHRs) and staff (NHS). Blood samples were collected from 115 NHRs and 254 NHS at baseline and 14 days after primary course BNT162b2 vaccination. Baseline samples were assessed for serum 25-hydroxyvitamin D levels, while follow-up samples were analyzed for spike protein S1 receptor-binding domain (S1RBD) IgG antibody concentrations and 50% pseudoneutralization titers. Vitamin D supplementation status was obtained from NHRs medical records. We compared immune responses between (severe) vitamin D-deficient and -sufficient NHRs/NHS and between supplemented and non-supplemented NHRs, stratified for history of SARS-CoV-2 infection and participant type. No significant differences in either binding or neutralizing COVID-19 vaccine antibody response were found between groups. The prevalence of vitamin D deficiency (<20 ng/mL) was 45% (95% CI: 36–54%) among NHRs and 60% (95% CI: 54–66%) among NHS. Although we showed that vitamin D status may not be related to a better COVID-19 vaccine antibody response, addressing the high prevalence of vitamin D deficiency in the nursing home population remains important.

## 1. Background

Vitamin D is a crucial nutrient involved in multiple physiological processes, including immunity [1,2,3,4,5,6,7,8]. By interacting with both adaptive (T-/B-cells) and innate immune cells (monocytes/dendritic cells), vitamin D can co-regulate immune cell differentiation and cytokine production, which is essential for effective immunogenicity against microbial infections [9,10]. When an individual’s 25-hydroxyvitamin D (25(OH)D) serum levels decrease below 20 ng/mL, we commonly speak of a pathological condition, referred to as vitamin D deficiency [11]. Previous studies have shown that vitamin D deficiency is associated with increased susceptibility to viral infections such as influenza, Respiratory Syncytial Virus (RSV), and Human Immunodeficiency Virus (HIV) [12,13,14]. Recently, different studies have shown that vitamin D may also play a role in protecting against coronavirus disease 19 (COVID-19), with higher levels of vitamin D and/or vitamin D supplementation being linked to better survival rates, less severe disease outcomes, and protection against infection [15,16,17,18,19,20,21,22,23]. In addition, a limited number of studies have investigated the association between vitamin D levels and COVID-19 vaccine-induced immune responses. Here, contrasting findings exist. Most studies report that vitamin D levels and/or supplementation are not associated with COVID-19 vaccine immunogenicity [24,25,26,27]. However, some report a positive association between vitamin D levels/supplementation and COVID-19 vaccine antibody responses [28,29].

While the relationship between vitamin D levels and COVID-19 vaccine immunogenicity has been investigated by a limited number of studies in healthy adults, little is known about this association in older adults, such as nursing home residents (NHRs). This population is particularly vulnerable to vitamin D deficiency, which is a major public health problem in the NHR population [30,31,32,33]. Moreover, older adults living in nursing homes (NHs) are known to exhibit impaired antibody responses to COVID-19 vaccination, compared to younger healthy individuals [34,35]. In light of this evidence, we conducted a study to investigate vitamin D status in nursing home residents and nursing home staff, 14 days after receiving two doses of the Pfizer/BioNTech BNT162b2 COVID-19 vaccine. Our aim was to determine whether (severe) vitamin D deficiency or treatment with vitamin D supplementation is associated with severe acute respiratory syndrome coronavirus 2 (SARS-CoV-2) antibody responses following vaccination.

## 2. Materials and Methods

### 2.1. Study Design and Population

The current study is a secondary analysis of data from a study that aimed to assess BNT162b2 vaccine immune response in nursing home residents, and has been described in detail elsewhere [35]. In brief, a total of 138 NHRs and 312 nursing home staff (NHS) from six Flemish NHs were recruited between 18 January and 4 March 2021. Venous blood was collected on the day of administration of the first BNT162b2 vaccine dose (baseline) and 14 days after administration of the second dose (follow-up). Baseline samples were used to assess history of SARS-CoV-2 infection and 25(OH)D levels, while follow-up samples were used to assess vaccine antibody response.

### 2.2. Ethics

This study was approved by the Ethical Committee of Ghent University Hospital (reference number BC-07665) and conducted according to the principles of the Declaration of Helsinki. Participants were informed about the study goals and procedures before written consent was obtained. A confidential counselor, such as a nurse, signed for participants who were incapable of signing the consent form, such as residents with dementia, whose consent was given by their legal representative.

### 2.3. Sample Collection

Approximately 5 mL of venous blood was obtained from each participant. Serum tubes were transported to the Laboratory of Clinical Microbiology of the Ghent University Hospital (Ghent, Belgium) within six hours after blood collection. There, serum tubes were centrifuged at 2000× *g* for 8 min and stored at 4 °C. The next day, serum was aliquoted and frozen at −20 °C until further analysis.

### 2.4. Antibody Detection

In the baseline samples, SARS-CoV-2 nucleocapsid protein (NCP)-specific immunoglobulin G (IgG)-binding antibodies were detected using a semi-quantitative enzyme-linked immune sorbent assay (ELISA) (EUROIMMUN, Lübeck, Germany) according to the manufacturer’s instructions. Results were expressed in optical density (OD) ratio. Samples that exceeded an OD ratio of 0.8 were considered seropositive, and therefore considered as previously SARS-CoV-2 infected. In the follow-up samples, SARS-CoV-2 spike protein S1 receptor-binding domain (S1RBD)-specific IgG-binding antibody concentrations were detected using a quantitative ELISA assay (ImmunoDiagnostics, Hong Kong), which we have previously validated [36]. Results were expressed in International Units/mL (IU/mL).

### 2.5. Pseudovirus Neutralization Assay

The neutralizing capacities of SARS-CoV-2 vaccine-induced antibodies were assessed using a pseudovirus (recombinant immunodeficiency virus 1 SG3ΔEnv) neutralization assay, as described before [35]. The highest antibody serum dilution resulting in 50% pseudovirus neutralization was reported as the 50% pseudovirus neutralization titer.

### 2.6. 25-Hydroxyvitamin D Detection and Supplementation

A metabolite of vitamin D, serum 25(OH)D, was measured using a quantitative ELISA assay with 6-point calibration (EUROIMMUN, Lübeck, Germany) according to the manufacturer’s instructions at baseline. The EUROIMMUN assay kit reliably detects 25(OH)D3 and 25(OH)D2. Vitamin D deficiency and severe vitamin D deficiency were defined as a 25(OH)D serum concentration below 20 ng/mL and 12 ng/mL, respectively [11,37]. Data on the use of vitamin D supplementation was obtained from the medical records for NHRs only.

### 2.7. Statistical Analysis

Differences in S1RBD IgG response and 50% pseudoneutralization titer (both post-vaccination) between vitamin-D-deficient and -sufficient participants were assessed using 20 ng/mL (deficiency) and 12ng/mL (severe deficiency) of serum 25(OH)D levels as cut-off. The analysis was stratified for participant type (NHRs/NHS) and infection status, as these factors were previously shown to affect vaccine antibody response [35]. Mann–Whitney U tests were performed to assess statistically significant differences (*p*-value ≤ 0.05) between groups (vitamin D sufficiency vs. deficiency and supplemented vs. non-supplemented). No multiplicity adjustments were carried out. Samples with a 50% pseudoneutralization titer of 0.00 were adjusted to a value of 0.01 in order to plot them on a logarithmic scale. The proportions of vitamin-D-deficient NHRs and NHS are reported with 95% confidence intervals (95% CI). All statistical analyses were performed in Graphpad Prism Version 9.3 (GraphPad Software, San Diego, CA, USA).

## 3. Results

### 3.1. Participant Characteristics

From all recruited participants (138 NHRs and 312 NHS), data from 115 NHRs and 254 NHS from five different NHs were included in the current analysis (participants with missing data regarding their age and/or information on vitamin D supplementation were excluded). Sociodemographic and participant characteristics of the study population are summarized in Table 1. The median age for NHRs was 89 years old, and 81% were female. The median age for NHS was 51 years old and 82% were female. A high prevalence of vitamin D deficiency was observed in our study population, with 45% (95% CI: 36–54%) of NHRs and 60% (95% CI: 54–66%) of NHS having serum 25(OH)D levels below 20 ng/mL. A total of 24% and 11% of NHRs and NHS, respectively, had severe vitamin D deficiency (<12 ng/mL). Median 25(OH)D levels and interquartile range was 20.80 ng/mL (16.85–24.75) and 18.19 ng/mL (14.94–23.02) for NHRs and NHS, respectively. Among NHRs, 37% (*n* = 42) received vitamin D supplementation. 25(OH)D levels were significantly higher in supplemented NHRs (27.97 ng/mL; 24.82–33.49), compared to non-supplemented NHRs (13.39 ng/mL; 10.82–22.42) (*p* < 0.0001).

### 3.2. COVID-19 Vaccine Binding and Neutralizing Antibody Responses in Vitamin-D-Deficient versus Sufficient NHRs and NHS

SARS-CoV-2 S1RBD IgG levels and 50% pseudoneutralization titers measured 14 days after the second dose of the BNT162b2 vaccine were compared between vitamin-D-deficient (<20 ng/mL) and vitamin D sufficient (≥20 ng/mL) participants, stratified per participant type and infection status. No significant differences in either binding or neutralizing antibody response were found between vitamin-D-deficient participants compared to participants with vitamin D sufficiency, not for residents nor for staff (Figure 1). In addition, when applying a cutoff of 12 ng/mL 25(OH)D serum concentration for severe vitamin D deficiency, no significant differences in binding/neutralizing antibody response between groups were observed (*p* > 0.05, Appendix A).

### 3.3. COVID-19 Vaccine Binding and Neutralizing Antibody Responses in Vitamin D Supplemented NHRs versus Non-Supplemented NHR

We compared post-vaccination S1RBD IgG antibody concentrations and 50% pseudoneutralization titers between vitamin D supplemented and non-supplemented NHRs. No significant differences in post-vaccination S1RBD IgG levels nor 50% pseudoneutralization titers were found between NHRs treated with vitamin D supplements compared to non-supplemented NHRs (Figure 2).

## 4. Discussion

In the current study, we aimed to investigate the potential association between vitamin D deficiency/vitamin D supplementation and SARS-CoV-2 antibody responses in NHRs and NHS following two doses of the BNT162b2 COVID-19 vaccine. Our findings show that neither (severe) vitamin D deficiency nor treatment with vitamin D supplementation were associated with antibody responses, either binding (S1RBD IgG) or neutralizing (50% pseudovirus neutralization titer), after COVID-19 vaccination in NHRs and NHS.

Our results are in line with what most other report concerning the association between vitamin D status, deficiency and/or supplementation and SARS-CoV-2 immune responses following vaccination in healthy adults. The majority of previous studies have similarly found no association between vitamin D and different immune endpoints following COVID-19 vaccination, such as binding antibodies, neutralizing antibodies, antibody decline and cellular responses [24,25,26,27]. Yet, a limited pair of studies report conflicting findings. One study found that vitamin D levels were positively associated with antibody response two weeks following a single BNT162b2 dose [28]. However, this study was limited by a small sample size and did not measure the antibody response after the two doses of the BNT162b2 vaccine, which is the recommended vaccine schedule. Moreover, their data show that 8 weeks following a single vaccine dose, differences in antibody response are no longer observed between the deficient/insufficient/replete groups. Another study identified regular intake of vitamin D supplementation as a predictor for SARS-CoV-2 seropositivity at a median of 8.6 weeks after COVID-19 vaccination in a population-based study [29]. Nevertheless, the association between use of vitamin D supplements and antibody levels as a continuous variable was not found there.

Supporting our data, other evidence can be found in studies investigating the association between vitamin D and antibody responses following vaccination against other viral infections, such as influenza. These studies have similarly demonstrated that neither vitamin D supplementation, vitamin D levels nor vitamin D deficiency, were associated with post-influenza vaccination antibody levels in both younger as older adults [38,39,40,41]. One study in particular, investigating the effect of vitamin D supplementation in nursing home residents, found that vitamin D supplementation did not affect antibody responses upon influenza vaccination, however, it did increase certain inflammatory cytokine levels post-vaccination [38].

A secondary interesting finding of the current study is the prevalence of vitamin D deficiency measured in our randomly recruited study population in Flanders, Belgium. We found that 45% (95% CI: 36–54%) of NHRs and 60% (95% CI: 54–66%) of NHS were vitamin D deficient (<20 ng/mL), and 24% (95% CI: 17–33%) of NHRs and 11% (95% CI: 7–15%) of NHS severe vitamin D deficient (<12 ng/mL). Other observational studies in a general European population have similarly reported a prevalence of vitamin D deficiency (<20 ng/mL) of ~40% and severe vitamin D deficiency (<12 ng/mL) of 13% [37,42]. In NHR populations, the prevalence of vitamin D deficiency is generally known to be even higher, with prevalence measured up to 94% in non-supplemented individuals [30]. One other study, dated from 2012, assessed the prevalence of vitamin D deficiency among NHRs from 53 NHs in Belgium, and reported a prevalence of 75.6% [43]. Multiple factors are known to affect vitamin D status in older adults, for example, poor daylight exposure, inadequate dietary intake, poor renal function, or reduced gut absorption due to ageing [30]. In our study population, however, 37% of NHRs were being treated with vitamin D supplements, which could explain the relatively low prevalence of vitamin D deficiency, compared to what others reported. Although we did not collect data on vitamin D supplementation in NHS, it could be speculated that a lower percentage of NHS were under vitamin D supplementation, which could explain the higher prevalence of vitamin D deficiency in NHS compared to NHRs. Regardless, Belgian guidelines recommend a daily intake of 800–2000 international units of vitamin D supplementation for all institutionalized elderly people [44]. Additionally, it is important to note that >80% of our study population were female, and the prevalence of vitamin D deficiency is known to be higher in women, with the highest difference observed between men and women in young adolescence and middle age [45]. Moreover, it should be noted that, although the observed prevalence of vitamin D deficiency was higher in NHS than NHRs, more NHRs had severe vitamin D deficiency.

Although vitamin D deficiency is highly prevalent among NHRs, and generally, older adults are known to have decreased immune responses following COVID-19 vaccination [30,34,35,46], our study suggests that vitamin D deficiency may not be the primary factor contributing to the observed impaired immune responses in nursing home residents following COVID-19 vaccination. In the literature, several studies have demonstrated that there is a negative association between vitamin D levels and COVID-19 severity in a pre-vaccination context [15,16,17,18,19,20], yet, our findings show that vitamin D is not associated with antibody levels two weeks after vaccination with the BNT162b2 COVID-19 vaccine. Most likely, other mechanisms exist through which vitamin D induces better COVID-19 outcomes. One study has suggested that vitamin D acts as a protecting agent by suppression of pro-inflammatory cytokines, which were shown to be negatively correlated with vitamin D levels [47]. Additionally, other mechanisms have been suggested by which vitamin D acts protectively against severe COVID-19, like induction of the transcription of the antimicrobial peptides cathelicidin and defensin [48]. Nevertheless, more fundamental research will be needed to fully understand how vitamin D status is associated with COVID-19 severity. Additionally, it is important to emphasize that, although vitamin D status may not be related to a better COVID-19 vaccine antibody response, it remains important to address the high prevalence of vitamin D deficiency among the nursing home population, as vitamin-D-deficient older adults are at high risk for multiple pathological conditions, e.g., cognitive decline, depression, osteoporosis, and increased risk of falling [49,50,51,52,53].

### Limitations

The current study assessed the association between vitamin D levels/supplementation and COVID-19 vaccine-induced antibody responses in a large sample of NHRs and NHS, however this study has limitations. Firstly, the results in this study were obtained by an observational study design, and not an experimental study design with controlled treatment groups. Moreover, we considered vitamin D blood levels as sufficient when 25(OH)D serum concentrations were 20 ng/mL or more. However, different classifications exist, as levels between 20 and 29 ng/mL are often categorized as insufficient, and those of 30 ng/mL or more as sufficient [8]. In our observations, not many participants had 25(OH)D levels that exceeded 30 ng/mL. Therefore, the effects 25(OH)D levels in the higher range of healthy blood levels, could not be investigated. Nevertheless, circulating serum 25(OH)D levels are known as a robust and reliable marker of vitamin D status [54]. Additionally, data on the use of vitamin D supplementation at baseline was obtained for NHRs from their medical record, however, no data were collected concerning the dosing and duration of supplementation.

## 5. Conclusions

Several studies have demonstrated a negative association between vitamin D status and COVID-19 severity in a pre-vaccination context. In the present study, we aimed to investigate whether (severe) vitamin D deficiency/treatment with vitamin D supplementation is associated with the SARS-CoV-2 antibody response measured 14 days after two doses of the BNT162b2 vaccine, in 115 NHRs and 254 NHS from 5 Belgian NHs. We found that neither (severe) vitamin D deficiency, nor vitamin D supplementation was associated with binding or neutralizing COVID-19 vaccine antibody response. Secondly, we found a high prevalence of vitamin D deficiency (<20 ng/mL) of 45% (95% CI: 36–54%) among NHRs and 60% (95% CI: 54–66%) among NHS, and severe vitamin D deficiency (<12 ng/mL) of 24% (95% CI: 17–33%) among NHRs and 11% (95% CI: 7–15%) among NHS. Although vitamin D status may not be related to a better COVID-19 vaccine antibody response, it remains important to address the high prevalence of vitamin D deficiency among the nursing home population, as vitamin-D-deficient older adults are at high risk for multiple other pathological conditions. More fundamental research will be needed to fully understand how vitamin D is associated with COVID-19 severity.

## Figures and Tables

**Figure 1 vaccines-11-01343-f001:**
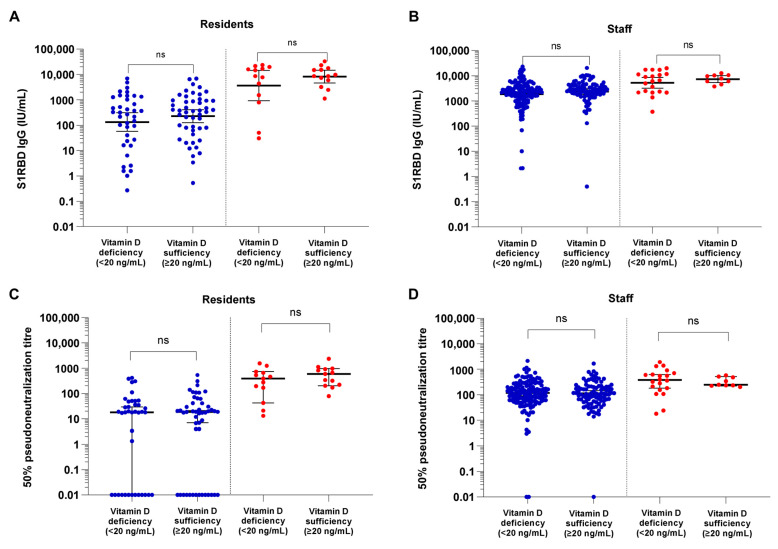
S1RBD IgG antibody concentration (International Unit/mL) (**A**,**B**) and 50% pseudoneutralization titer (**C**,**D**) 14 days after BNT162b2 vaccination among 25-OH vitamin-D-deficient (<20 ng/mL) and 25-OH vitamin D sufficient (≥20 ng/mL) nursing home residents (**A**,**C**) and nursing home staff (**B**,**D**). Data is presented stratified for infection naïve participants (blue) and previously infected participants (red). Bold horizontal lines with error bars represent the geometric mean S1RBD IgG antibody concentration and median 50% pseudoneutralization titer per group with 95% confidence intervals. ns: not significant at the 0.05 level. Samples with a 50% pseudoneutralization titer of 0.00 were adjusted to a value of 0.01 in order to plot them on a logarithmic scale.

**Figure 2 vaccines-11-01343-f002:**
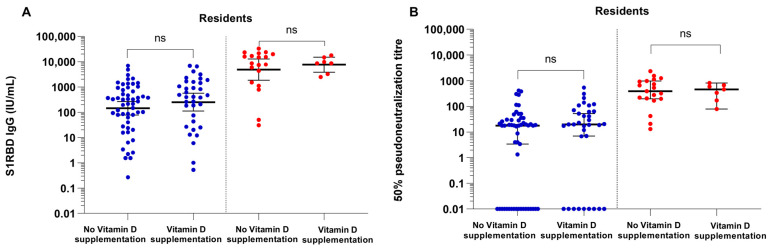
S1RBD IgG antibody concentrations (International Unit/mL) (**A**) and 50% pseudoneutralization titers (**B**) 14 days after BNT162b2 vaccination among vitamin D supplemented and non-supplemented nursing home residents. Data is presented stratified for infection naïve residents (blue) and previously infected residents (red). Bold horizontal lines with error bars represent the geometric mean S1RBD IgG antibody concentration and median 50% pseudoneutralization titer per group with 95% confidence intervals. ns: not significant at the 0.05 level. Samples with a 50% pseudoneutralization titer of 0.00 were adjusted to a value of 0.01 in order to plot them on a logarithmic scale.

**Table 1 vaccines-11-01343-t001:** Participant characteristics of nursing home residents and staff.

	Nursing Home Residents (*n* = 115)	Nursing Home Staff (*n* = 254)
**Sociodemographic characteristics**		
Female, *n* (%)	93 (81%)	208 (82%)
Age, median (interquartile range)	89 (86–93)	51 (36–63)
**25(OH)D**		
25(OH)D serum concentration (ng/mL), median (interquartile range)	20.80 (16.85–24.75)	18.19 (14.94–23.02)
25(OH)D serum concentration non-supplemented (ng/mL), median (interquartile range)	13.39 (10.82–22.42)	NA
25(OH)D serum concentration supplemented (ng/mL), median (interquartile range)	27.97 (24.82–33.49)	NA
25(OH)D serum concentration previous SARS-CoV-2 infection (ng/mL), median (interquartile range)	19.71 (11.52–28.29)	18.52 (16.90–20.82)
25(OH)D serum concentration no history of SARS-CoV-2 infection (ng/mL), median (interquartile range)	22.56 (13.13–26.69)	18.22 (14.94–23.26)
25-OH Vitamin D deficiency (<20 ng/mL), *n* (%; 95% CI)	52 (45%; 36–54%)	153 (60%; 54–66%)
25-OH Vitamin D severe deficiency (<12 ng/mL), *n* (%; 95% CI)	28 (24%; 17–33%)	27 (11%; 7–15%)
Vitamin D supplementation, *n* (%)	42 (37%)	NA
**SARS-CoV-2**		
Previous SARS-CoV-2 infection, *n* (%) ^a^	26 (23%)	29 (11%)
SARS-CoV-2 IgG concentration post-vaccination, geometric mean (interquartile range)	388.24 (82.15–2237.43)	2286.64 (1630.52–4521.25)
SARS-CoV-2 50% pseudoneutralization titer, median (interquartile range)	20.96 (2.35–153.3)	124.00 (61.71–271.48)
Prevalence seroconversion after vaccination, *n* (%)	102 (89%)	250 (98%)

^a^ Assessed by detection of anti-nucleocapsid protein IgG in baseline samples.

## Data Availability

The data presented in this study are available on request from the corresponding authors.

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
