# Peer review of "No Significant Association between 25-OH Vitamin D Status and SARS-CoV-2 Antibody Response after COVID-19 Vaccination in Nursing Home Residents and Staff"

_vaccines, 2023, doi:10.3390/vaccines11081343_

Round 1
Reviewer 1 Report
The authors aimed to investigate 21 the association between vitamin D deficiency/supplementation, and SARS-CoV-2 antibody responses post-vaccination in nursing home residents (NHR) and staff (NHS). It is a confirmatory study. Others have investigated the same topic. Their results are in line with what most others have reported concerning the association between vitamin D level, deficiency and/or supplementation and SARS-CoV-2 immune responses following vaccination in healthy adults. The methodology use was correct. They divided the participants according to whether they supplemented with vitamin D or not. Additionally, data on the use of vitamin D supplementation at baseline was obtained for NHR from their medical record, however, no data was collected concerning the dosing and duration of supplementation. Additional data might have been useful. However, given the findings showing a large difference between those supplemented and those not supplemented, it was not needed.The conclusions are consistent with the evidence and arguments presented and they address the main question posed. In addition, this is an important point: Although we showed that vitamin D status may not be related to a better COVID-19 vaccine antibody response, addressing the high prevalence of deficiency in nursing home populations remains important.
Tables and figures are fine.
The analysis seems fine. My comments are related to the choice of references.
|
[47] R. Vieth, “What is the optimal vitamin D status for health?,” Progress in Biophysics and Molecular Biology, vol. 92, no. 1, pp. 26– |
405 |
|
32, 2006/09/01/ 2006, doi: https://doi.org/10.1016/j.pbiomolbio.2006.02.003. [48] B. Dawson-Hughes et al., “IOF position statement: vitamin D recommendations for older adults,” Osteoporosis International, vol. |
406407 |
|
21, no. 7, pp. 1151–1154, 2010/07/01 2010, doi: 10.1007/s00198–010-1285–3. |
40 |
Comment: These and many papers among the first twelve are too old and should be replaced by recent ones. Try to find papers from the past 2-4 years. Google Scholar is a good place to search.
Some suggestions:
Disassociation of Vitamin D's Calcemic Activity and Non-calcemic Genomic Activity and Individual Responsiveness: A Randomized Controlled Double-Blind Clinical Trial.
Sci Rep. 2019 Nov 27;9(1):17685. doi: 10.1038/s41598-019-53864-1.
Vitamin D for skeletal and non-skeletal health: What we should know.
J Clin Orthop Trauma. 2019 Nov-Dec;10(6):1082-1093. doi: 10.1016/j.jcot.2019.07.004
Immunologic Effects of Vitamin D on Human Health and Disease.
Nutrients. 2020 Jul 15;12(7):2097. doi: 10.3390/nu12072097.
The Implications of Vitamin D Status During Pregnancy on Mother and her Developing Child.
Front Endocrinol (Lausanne). 2018 Aug 31;9:500. doi: 10.3389/fendo.2018.00500.
or
Effectiveness of Prenatal Vitamin D Deficiency Screening and Treatment Program: A Stratified Randomized Field Trial.
J Clin Endocrinol Metab. 2018 Aug 1;103(8):2936-2948. doi: 10.1210/jc.2018-00109.
Association between vitamin D supplementation and COVID-19 infection and mortality.
Sci Rep. 2022 Nov 12;12(1):19397. doi: 10.1038/s41598-022-24053-4.
Vitamin D supplementation to prevent acute respiratory tract infections: systematic review and meta-analysis of individual participant data.
BMJ. 2017 Feb 15;356:i6583. doi: 10.1136/bmj.i6583.
Additional references re SARS-CoV-2 antibodies found by searching Google Scholar. Please check there for more.
Determinants of pre-vaccination antibody responses to SARS-CoV-2: a population-based longitudinal study (COVIDENCE UK)
M Talaei, S Faustini, H Holt… - BMC …, 2022 - bmcmedicine.biomedcentral.com
SARS‐CoV‐2 specific antibody responses in healthcare workers after a third booster dose of CoronaVac or BNT162b2 vaccine
E Yavuz, Ö Günal, E Başbulut… - Journal of Medical …, 2022 - Wiley Online Library
Author Response
We thank the Reviewer for his/her comment and for suggesting manuscripts to cite. We have verified the used references in the manuscript once more and added some additional, more recent, sources from your suggestions and found in literature. However, we believe that, besides the date of publishment, it is of equal or higher importance that the cited source strengthens and/or justifies what is written in the text and originates from a reliable source. Therefore, we decided not remove any sources.
Reviewer 2 Report
Major comments:
1. Please be more specific, which compound is meant, when you use the term "Vitamin D". Is it vitamin D3, 25(OH)D3 or 1,25(25)2D3?
2. Is it really correct to compare NH residents with staff, since the average age is significantly different? It is know that the immune system declines with age. Please discuss this more detailed. Moreover, it is interesting that the staff has even worse vitamin D levels than the residents. Also this should be discussed.
3. it is know that people can be high, mid and low responders to vitamin D. This distinction may be more important than the concrete vitamin D status. Please discuss.
Minor comments:
1. Please define all abbreviations at their first time use and apply them then consistently.
2. Please have the table better integrated into the text, i.e. not on two pages.
Author Response
Major comments:
- Please be more specific, which compound is meant, when you use the term "Vitamin D". Is it vitamin D3, 25(OH)D3 or 1,25(25)2D3?
We have added the vitamin D form in the methods section, line 113, and has been updated in the abstract, line 22. The EUROIMMUN 25-OH Vitamin D ELISA kit detecting 25(OH)D3 and 25(OH)D2 was used in the study.
- Is it really correct to compare NH residents with staff, since the average age is significantly different? It is know that the immune system declines with age. Please discuss this more detailed. Moreover, it is interesting that the staff has even worse vitamin D levels than the residents. Also this should be discussed.
We agree with the reviewer that NH residents and staff are two completely different age groups which can affect immune function. However, we would like to emphasize that we are not comparing the two groups in this manuscript. Instead, we stratified the analysis per NH residents and staff, as we previously showed that the immune response upon COVID-19 vaccination is different in both groups [1-3]. We are only comparing vitamin D deficient vs. sufficient and supplemented vs non-supplemented participants within the strata to control for the age effect. To make this clearer, we added some details in line 123-124 regarding the statistical analysis, as this might have been confusing.
- it is know that people can be high, mid and low responders to vitamin D. This distinction may be more important than the concrete vitamin D status. Please discuss.
We acknowledge that literature has demonstrated that there is a large individual variation in the response to vitamin D supplementation. However, in our study population, we showed a large difference in serum 25(OH)-D concentrations between supplemented and non-supplemented nursing home residents (see Table 1). We therefore believe that the inter-individual variance in response to supplement intake does not affect our findings.
Minor comments:
- Please define all abbreviations at their first time use and apply them then consistently.
We have verified the complete manuscript on the use of abbreviations.
- Please have the table better integrated into the text, i.e. not on two pages.
We confirm that this issue will be solved during the proofreading step.
Reviewer 3 Report
This MS reports a study of antibody responses to one specific Covid-19 vaccine in staff and residents of a number of nursing homes aimed at determining whether the antibody responses to a that Covid-19 vaccine might be improved by higher vitamin D status. The study revealed vitamin D deficiency rates of 45% and 60% in patients and staff, respectively, but found no variations in mean 2-week antibody responses to vaccination either with baseline vitamin D status or a with a record of supplementation in either group. The report is quite easy to understand though some minor edits of the English language would increase its readability as listed in the specific [minor] comments.
General [major] comments. [By line number]
- Line 105, vitamin D was not measured, it was, as the authors clearly report, the 25-hydroxyvitamin D metabolite, and it would be helpful to readers to use that term in this heading. Line 106+ 25(OH)D assays are liable to much variability . It is, therefore, usual to report both the QC scheme used to validate these assays [often the international DEQAS scheme] and that the laboratory concerned was in compliance with the standards of that QC scheme, together with the cumulative variance [CV] of the assay data. This information could easily be provided by the laboratory concerned.
- Similarly, where results of ‘vitamin D levels’ are reported they are actually ‘25(OH)D concentrations’. It would be better to use that term in the Results but if not used more generally then you might use the term ‘vitamin D status’, already used in the title, instead of ‘vitamin D levels’, in other sections such as the Discussion.
3. A plot of the various measures of antibody response against actual serum 25(OH)D values, could prove to be of interest since variation in those responses with variation in vitamin D status may be lost when means/medians are used for comparison, for example there might be skews in the distributions or plateaus of biological significance that are not detectable when comparing means/medians. It would, therefore, be worth looking at this form of data display and reporting on it in the staff and patient groups separately and in the supplemented and unsupplemented patients , despite the relatively small numbers and the authors would need to consider anything notable that might emerge.
4. Line 34, The authors emphasize that the highly prevalent deficiency seen in nursing homes needs to be addressed but, since this was even more prevalent in staff than in patients, this point should be mentioned here in the Abstract as well as being made in the main MS, especially since women made up > 80% of each study group.
Specific [minor] comments [relating to clarity or to English usage, by line number].
Line 50, please separate out those refs that were ‘prospective, [including the study by Kaufmann et al. which was very large, looked at D status many months before the pandemic began and was well adjusted for other risk factors and for season] from those made during covid-19 illness [amongst refs 13-20]. Line 54 might be a useful place to say “serum 25(OH)D concentration [vitamin D status]” so that you can use that term later in the text, e.g. Line 76, Line 105, etc, without needing to qualify it. Line 65, if there really is no clue to how much vitamin D residents were given, it must have been pretty small amount for 40+% of residents to remain deficient and perhaps stating the Belgian recommendations for elders here would , therefore, be useful as my bet is that that amount is also rather small. Line 115, since serum 25(OH)D thresholds vary for different effects have you checked that using a higher threshold, such as the 30ng/ml [75 nmol/l] used to define repletion by the American Endocrine Society does not reveal any differences in the 2-week responses to vaccination? Line ~137, were the 25(OH)D values of supplemented residents as high as those of replete staff I wonder, since staff were younger and responses to VitD intakes are greater in younger versus older adults. It is possible that a comparison of the histograms of distribution of 25(OH)D values in residents vs staff and in treated and untreated residents vs staff might be useful?? Line 150, typo, 25-hydroxyvitamin D…… Lines 172/3, there is no colour in the file of this MS, so that I cannot see this feature or what it may suggest if anything? Line 191, in reporting another study [25] as showing a difference in antibody response by Vit D status please state the time interval involved since, if it was longer than the 2 weeks in the present study then this might make the choice of 2 week sampling a weakness. Line 203, …study in particular would be better than in specific. Line 209, …of NHS were… not was. Line 218, say either, ‘…. in the elderly’, or ‘……in elders’, wherever this term is used. Line 227, ‘In the literature…’ Line 229. That vitamin D status was not ….. 2 weeks after vaccination with the xxxx vaccine, ……… Line 231, induces would be more correct han warrants. After that sentence there is much more work showing how better D status should be protective for Covid-19 severity, and you might, therefore, like to add something to that section, [possibly one of the 2 recent papers from Griffin G + Hewison M and others]. Line 244, …vitamin D status, taking vitamin D supplements …; [since there are many other types of supplements], line 252, …level is known as …. [or you can say ‘ … levels are known as ….. ] …. Line 257, …vitamin D status and [since many readers only look at ‘Conclusions’].
The English is pretty good but there are several places where it needs to be tweaked to make the text easier to follow or easier to read; these are noted in the specific [minor] comments to the authors
Author Response
We thank the reviewer to take the time to thoroughly review our manuscript.
General [major] comments. [By line number]
- Line 105, vitamin D was not measured, it was, as the authors clearly report, the 25-hydroxyvitamin D metabolite, and it would be helpful to readers to use that term in this heading.
We adjusted the heading as such.
- Line 106+ 25(OH)D assays are liable to much variability . It is, therefore, usual to report both the QC scheme used to validate these assays [often the international DEQAS scheme] and that the laboratory concerned was in compliance with the standards of that QC scheme, together with the cumulative variance [CV] of the assay data. This information could easily be provided by the laboratory concerned.
As the current analyses were conducted in a research laboratory, there was no participation in an international DEQAS quality control scheme. However, for assessment of 25(OH)D concentrations, we used a well-established commercial assay (EUROIMMUN, Lübeck, Germany) with a reported CV between 3.2% and 8.6% (see assay brochure).
- Similarly, where results of ‘vitamin D levels’ are reported they are actually ‘25(OH)D concentrations’. It would be better to use that term in the Results but if not used more generally then you might use the term ‘vitamin D status’, already used in the title, instead of ‘vitamin D levels’, in other sections such as the Discussion.
We agree with the reviewer that use of different terminology can create confusion. Therefore, we replaced ‘vitamin D levels’ by 25(OH)D levels/concentration when it concerned reporting our own results. For citing results from other articles, we have kept the term ‘vitamin d’ since different articles measure different metabolites.
- A plot of the various measures of antibody response against actual serum 25(OH)D values, could prove to be of interest since variation in those responses with variation in vitamin D status may be lost when means/medians are used for comparison, for example there might be skews in the distributions or plateaus of biological significance that are not detectable when comparing means/medians. It would, therefore, be worth looking at this form of data display and reporting on it in the staff and patient groups separately and in the supplemented and unsupplemented patients , despite the relatively small numbers and the authors would need to consider anything notable that might emerge.
During exploratory analysis, we plotted the continuous 25(OH)D data versus the continuous binding and neutralizing levels. No trends were visible in these exploratory graphs (see Figure 1 in attached file). We decided to not include this figure in the manuscript, since it would distract the attention from our main graphs.
Reviewer 4 Report
The communication under review investigates the potential link between vitamin D status, COVID-19 vaccine antibody response, and the prevalence of vitamin D deficiency among the nursing home population. Although the topic is timely, the overall merit of the study are rather low.
The title should be shortened.
While acknowledging the absence of a direct association between vitamin D status and COVID-19 vaccine antibody response, the authors emphasize the importance of addressing vitamin D deficiency in the elderly due to its association with multiple pathological conditions. They also highlight the need for further research to comprehensively understand the relationship between vitamin D and COVID-19 severity.
The data on vitamin D levels presented in the discussion sections should be moved to the results section. Nevertheless, the absence of relevant findings weakens the study's conclusions. Furthermore, the authors' assertion that addressing vitamin D deficiency remains important solely based on the high prevalence of deficiency (which is known to be very common in the entire population) among the nursing home population is weak. The manuscript does not offer any specific evidence or analysis to establish a causal link between vitamin D deficiency and other pathological conditions.
The limitations section does not include aspects of the actual topic of vitamin D and COVID-19 severity. So, while the paper acknowledges the need for further research, it fails to address the limitations of the study. The absence of a comprehensive discussion on potential confounding factors, the lack of a control group, and the exclusion of relevant variables weaken the paper's scientific rigor. Additionally, the absence of longitudinal data or follow-up studies limits the authors' ability to make definitive claims about the long-term implications of vitamin D deficiency and COVID-19 severity.
The tables and figures need some editing to imrove readability.
The conclusion should not include references.
Moderate editing of English language required
Author Response
The communication under review investigates the potential link between vitamin D status, COVID-19 vaccine antibody response, and the prevalence of vitamin D deficiency among the nursing home population. Although the topic is timely, the overall merit of the study are rather low.
We appreciate the Reviewer's effort to read and review the manuscript. However, we would like to highlight the scientific value of the current manuscript and convince the Reviewer of the manuscript’s importance.
We decided to conduct this study, as conflicting findings exist in literature concerning the association between vitamin D deficiency and COVID-19 antibody response. Therefore, and for the interest of stakeholders (e.g. clinicians, nursing home staff), we conducted this observational study as a follow-up of a previously executed study on COVID-19 antibody response among nursing home residents and staff. To our knowledge, we are first to investigate this association in nursing home residents, who are prone to be vitamin D deficient and have lower antibody response after COVID-19 vaccination. Moreover, our study assessed the prevalence of vitamin D deficiency in a randomly recruited selection of nursing home residents and staff from five different nursing homes in Flanders, Belgium (regardless of supplement intake). Other than that, the latest data on the prevalence of vitamin D deficient elderly in Belgian nursing home residents found in scientific literature dates from 2012 [4].
The title should be shortened.
We have changed the title into ‘No significant association between vitamin D status and SARS-CoV-2 antibody response after COVID-19 vaccination in nursing home residents and staff’ as suggested by the responsible Editor. However, we would be open to consider shorter suggestions.
While acknowledging the absence of a direct association between vitamin D status and COVID-19 vaccine antibody response, the authors emphasize the importance of addressing vitamin D deficiency in the elderly due to its association with multiple pathological conditions. They also highlight the need for further research to comprehensively understand the relationship between vitamin D and COVID-19 severity.The data on vitamin D levels presented in the discussion sections should be moved to the results section. Nevertheless, the absence of relevant findings weakens the study's conclusions. Furthermore, the authors' assertion that addressing vitamin D deficiency remains important solely based on the high prevalence of deficiency (which is known to be very common in the entire population) among the nursing home population is weak. The manuscript does not offer any specific evidence or analysis to establish a causal link between vitamin D deficiency and other pathological conditions.
We, as a multidisciplinary team of researchers and clinicians, believe that, although it is not the main conclusion of our paper, it remains an important side note that vitamin D deficiency in the nursing home population should be addressed. Multiple others have previously shown that vitamin D deficient elderly have an increased risk for multiple pathological conditions. We have added additional sources to strengthen this statement.
The limitations section does not include aspects of the actual topic of vitamin D and COVID-19 severity. So, while the paper acknowledges the need for further research, it fails to address the limitations of the study. The absence of a comprehensive discussion on potential confounding factors, the lack of a control group, and the exclusion of relevant variables weaken the paper's scientific rigor. Additionally, the absence of longitudinal data or follow-up studies limits the authors' ability to make definitive claims about the long-term implications of vitamin D deficiency and COVID-19 severity.
We would like to emphasize that the aim of the study is not to investigate the association between vitamin D and COVID-19 severity, but between vitamin D and COVID-19 antibody response. We therefore do not consider this as a limitation.
Moreover, we agree with the Reviewer that it is important to stress that the current findings were obtained in an observational study design, and not an randomized control trial. This was added to the Limitations section (line 267-268).
Additionally, in line with the comments of other reviewers, we have added more nuance to the discussion section (e.g. regarding longitudinal data and the high prevalence of vitamin D deficiency in the NH population) to the discussion section (see line 200-208 and line 233-242). We would also like to note that this article is submitted as a communication and we are therefore limited in words.
The tables and figures need some editing to imrove readability.
As no other comments were made regarding the readability of the tables/figures, we would be happy to hear what exactly should be changed to enhance their readability, and adapt them as such.
The conclusion should not include references.
The references from the conclusion section were removed.
Reviewer 5 Report
The authors investigated vitamin D levels' association with antibody responses from COVID-19 vaccination in nursing home residents and staff. Findings did not support an association of immune antibody response from vaccination with vitamin D deficiency or supplementation. Nevertheless, vitamin D levels are low in nursing home residents, and more research is needed to understand how vitamin D is linked to lower COVID-19 severity in older adults.
The authors have previously published a study on immune response from vaccination in nursing home residents. Results of the present study contribute to the literature finding no association of vitamin D deficiency or supplementation with antibody responses from COVID-19 vaccination.
I have a few more comments in the attached pdf file.

Author Response
The authors investigated vitamin D levels' association with antibody responses from COVID-19 vaccination in nursing home residents and staff. Findings did not support an association of immune antibody response from vaccination with vitamin D deficiency or supplementation. Nevertheless, vitamin D levels are low in nursing home residents, and more research is needed to understand how vitamin D is linked to lower COVID-19 severity in older adults.
The authors have previously published a study on immune response from vaccination in nursing home residents. Results of the present study contribute to the literature finding no association of vitamin D deficiency or supplementation with antibody responses from COVID-19 vaccination.
I have a few more comments in the attached pdf file.
We thank the reviewer for his/her effort to improve the manuscript and confirm the suggested changes were adapted.
Round 2
Reviewer 2 Report
The term "vitamin D" is still used at some occasion without clarification which metabolite is meant.
OK
Author Response
We thank the reviewer for the comment. We have now added the metabolite description in the title to avoid any possible confusion.
Reviewer 3 Report
The 'major comments' have been addressed adequately. However, the 'Specific [minor] comments' made, including those about looking for a higher threshold at which vaccine responses might be increased, perhaps at 30 ng/ml rather than the 20 ng/ml cutoff reported, about mentioning why higher vitamin D status is of importance for immune defences, [by quoting at least one mechanistic paper], and about discussing why nursing home staff had higher deficiency rates than nursing home residents have been ignored altogether as far as I can see.
The English is pretty good but there are several places where it needs to be tweaked to make the text easier to follow or easier to read; these are noted in the specific [minor] comments to the authors
Author Response
We thank the reviewer for the comment. No response to the minor comments was simply a copy-paste mistake for which we apologize. The missing text is now shown below:
Line 50, please separate out those refs that were ‘prospective, [including the study by Kaufmann et al. which was very large, looked at D status many months before the pandemic began and was well adjusted for other risk factors and for season] from those made during covid-19 illness [amongst refs 13-20].
We are not completely sure what is meant by this comment, as all cited references (13-20) are prospective/retrospective studies or meta-analyses/reviews. However, when clarified, we would be happy to make adaptations as such.
Line 54 might be a useful place to say “serum 25(OH)D concentration [vitamin D status]” so that you can use that term later in the text, e.g. Line 76, Line 105, etc, without needing to qualify it.
In line with the comment of Reviewer 2, we replaced ‘vitamin D levels’ by ‘25(OH)D levels/concentration’ throughout the manuscript when it concerned reporting our own results, to avoid the use of different terminology.
Line 65, if there really is no clue to how much vitamin D residents were given, it must have been pretty small amount for 40+% of residents to remain deficient and perhaps stating the Belgian recommendations for elders here would , therefore, be useful as my bet is that that amount is also rather small.
We thank the Reviewer for this thoughtful comment, however, we would like to note that 37% of residents in our study population (as shown in Table 1) were vitamin D supplemented (data collected from their medical records). Nevertheless, we did not collect any data on the dosing of the supplementation. The Belgian guidelines advise daily intake of vitamin D supplements of 800 to 2000 international units for every institutionalized elderly. We added this information in line 235-237.
Line 115, since serum 25(OH)D thresholds vary for different effects have you checked that using a higher threshold, such as the 30ng/ml [75 nmol/l] used to define repletion by the American Endocrine Society does not reveal any differences in the 2-week responses to vaccination?
In our observations, not many participants had 25(OH)D levels that exceeded 30ng/mL. Therefore, the effects 25(OH)D levels in the higher range of healthy blood levels, could not be investigated. This is discussed in the Limitations section.
Line ~137, were the 25(OH)D values of supplemented residents as high as those of replete staff I wonder, since staff were younger and responses to VitD intakes are greater in younger versus older adults. It is possible that a comparison of the histograms of distribution of 25(OH)D values in residents vs staff and in treated and untreated residents vs staff might be useful??
Unfortunately, we did not collect data on the use of vitamin D supplementation among NHS. The comparison between supplemented vs. non-supplemented participants was only done for NHR and it is therefore not applicable to include a comparison for 25(OH)D values in treated and untreated residents vs staff. Alternatively, we did present the mean 25(OH)D serum concentration in supplemented and non-supplemented in Table 1 to confirm the response to supplement intake.
Line 150, typo, 25-hydroxyvitamin D……
Adjusted.
Lines 172/3, there is no colour in the file of this MS, so that I cannot see this feature or what it may suggest if anything?
We confirm that the article figures will be published in color.
Line 191, in reporting another study [25] as showing a difference in antibody response by Vit D status please state the time interval involved since, if it was longer than the 2 weeks in the present study then this might make the choice of 2 week sampling a weakness.
We thank the reviewer for this interesting comment. We looked into this and added more details concerning the time of antibody measurement post-vaccination for the two papers reporting conflicting results from line 195-205.
Line 203, …study in particular would be better than in specific. Line 209, …of NHS were… not was. Line 218, say either, ‘…. in the elderly’, or ‘……in elders’, wherever this term is used. Line 227, ‘In the literature…’ Line 229. That vitamin D status was not ….. 2 weeks after vaccination with the xxxx vaccine, ……… Line 231, induces would be more correct han warrants.
Adjusted.
After that sentence there is much more work showing how better D status should be protective for Covid-19 severity, and you might, therefore, like to add something to that section, [possibly one of the 2 recent papers from Griffin G + Hewison M and others]. Line 244, …vitamin D status, taking vitamin D supplements …; [since there are many other types of supplements], line 252, …level is known as …. [or you can say ‘ … levels are known as ….. ] …. Line 257, …vitamin D status and [since many readers only look at ‘Conclusions’].
Adjusted
Reviewer 4 Report
I thank the authors for their response and correction of their manuscript. Still, I think that the article should undergo editing to comply with the jounrnal guidelines, e.g. regarding font size, figures and tables.
Moderate editing of English language as well as formatting required
Author Response
We thank the reviewer for the comment. We have checked the manuscript side by side with the guidelines. font types and sizes were adjusted where necessary, Table 1 has been reformatted.
Round 3
Reviewer 3 Report
I am satisfied that the points raised have been adequately addressed [so far as is possible with the data available and that those that could not be addressed due to lack of information are now clarified in the text and I have no further concerns to raise.